# Genetic and Serum Screening for Alpha-1-Antitrypsin Deficiency in Adult Patients with Cystic Fibrosis: A Single-Center Experience

**DOI:** 10.3390/biomedicines10123248

**Published:** 2022-12-14

**Authors:** Francesco Amati, Andrea Gramegna, Martina Contarini, Anna Stainer, Cristina Curcio, Stefano Aliberti, Angelo Guido Corsico, Francesco Blasi

**Affiliations:** 1Department of Biomedical Sciences, Humanitas University, Via Rita Levi Montalcini 4, 20072 Milan, Italy; 2IRCCS Humanitas Research Hospital, Respiratory Unit, Via Manzoni 56, 20089 Milan, Italy; 3Department of Pathophysiology and Transplantation, University of Milan, 20122 Milan, Italy; 4Internal Medicine Department, Respiratory Unit and Cystic Fibrosis, Fondazione IRCCS Ca’ Granda Ospedale Maggiore Policlinico, Via Francesco Sforza 35, 20122 Milan, Italy; 5Medical Genetics Laboratory, Fondazione IRCCS Ca’ Granda Ospedale Maggiore Policlinico, 20122 Milan, Italy; 6Center for Diagnosis of Inherited Alpha1-Antitrypsin Deficiency, Department of Internal Medicine and Therapeutics, University of Pavia, 27100 Pavia, Italy; 7Pneumology Unit IRCCS San Matteo Hospital Foundation, 27100 Pavia, Italy

**Keywords:** cystic fibrosis, alpha-1 antitrypsin deficiency, airway inflammation, bronchiectasis

## Abstract

Cystic fibrosis (CF) and alpha-1 antitrypsin (AAT) deficiency are two of the commonest genetic diseases affecting the Caucasian population. Neutrophil-mediated inflammation due to protease–antiprotease imbalance leads to progressive pulmonary involvement in both diseases. The aim of this study was to investigate the prevalence of AAT deficiency in CF adults. A prospective study enrolling CF adults was conducted at the Adult CF Center based in Milan from January 2018 to March 2019. Patients were tested for AAT serum protein quantification and expanded genotyping characterization of SERPINA1 during clinical stability. Genotyping characterization of SERPIN1 was compared to a control population of 2848 Caucasian individuals with the same geographical origin and similar demographic characteristics. Among 173 patients included in the study, the prevalence of AAT deficiency was 0. Genotype analysis was piMM in 166 (94.9%) patients and piMS in 9 patients (5.1%), respectively. No differences in terms of genotype characterization were found between the CF population and the control population. These data show that AAT deficiency is not common among adults with CF.

## 1. Introduction

Cystic fibrosis (CF) and alpha-1 antitrypsin (AAT) deficiency are two of the commonest progressive genetic diseases affecting the Caucasian population [1,2]. Lung damage in both these conditions is driven by neutrophil-mediated inflammation due to protease–antiprotease imbalance, leading to progressive pulmonary destruction [3]. CF lung disease can occur very early in life and is associated with early bacterial colonization and bronchiectasis appearance, whereas the airway disease in AAT deficiency is classically parenchymal, presenting usually by the third or fourth decade of life. Indeed, the clinical impact of AAT deficiency is highly heterogeneous in lung disease, being only partly explained by exposure to known risk factors. Several reports have suggested an association between AAT deficiency and bronchiectasis, particularly in patients with severe AAT deficiency (PiZZ phenotype) [4]. Furthermore, investigations of lung inflammation in CF have demonstrated an overwhelming burden of neutrophilic mediators in both bronchoalveolar lavage (BAL) and sputum [5]. The coexistence of both CF and AAT deficiency lung diseases can sustain a vicious vortex of airway inflammation, infection, and structural damage, leading to negative outcomes [6]. AAT deficiency is one of the commonest hereditary disorders among Caucasians, and its prevalence has been reported to vary greatly across regions. However, no extensive studies evaluating this hypothetical contributor of disease severity in CF have been conducted in Southern Europe. Thus, the aim of this study was to investigate the prevalence of AAT deficiency, both in terms of AAT serum levels and prevalence of deficient alleles, in a large cohort of adults with CF from Italy.

## 2. Materials and Methods

### 2.1. Study Design and Population

An observational, prospective, consecutive study was conducted on adults (≥18 years of age) in clinical follow-up at the Adult CF Center, Policlinico University Hospital in Milan, Italy. Consecutive CF outpatients were included in the study from January 2018 to March 2019. All participants provided written informed consent. The study received no funding. Patients with an ongoing respiratory exacerbation were excluded due to the evidence that AAT is an acute-phase protein that increases during acute infection states. Patients were tested for AAT serum protein quantification and expanded genotyping characterization of SERPIN1. Serum AAT was classified as low (<80 mcg/dL) or normal (≥80 mcg/dL), measured by radial immunodiffusion according to local standard operating procedures. Genotyping characterization of SERPIN1 was compared to a control population of 2848 Caucasian individuals with the same geographical origin and similar demographic characteristics.

### 2.2. Study Definitions

AAT deficiency was defined as both low serum level of AAT and a deficient genotype by European Respiratory Society (ERS) diagnostic criteria [7]. CF was defined according to Farrell and colleagues [1]. Chronic bacterial infection was defined as the presence of 2 cultures positive for pathogenic bacteria at least 3 months apart over 12 months [8]. Chronic *P. aeruginosa* infection was defined as the presence of >50% of sputum cultures being *P. aeruginosa* positive in the preceding 12 months [9]. Murray–Washington criteria for sputum quality were adopted, with all samples having <10 squamous cells and >25 leukocytes per low-power microscope field. Exacerbation of CF was defined as a deterioration in symptoms causing a physician to change treatments [10].

### 2.3. Study Endpoints

The primary endpoint was to determine the prevalence of patients with AAT deficiency in a large CF cohort from Southern Europe. Secondary endpoints explored the prevalence and clinical characteristics of patients with S-allele and Z-allele heterozygosity.

### 2.4. Study Groups

Patients were grouped according to the results of the conducted SERPIN-1 genetic test and divided into deficient allele heterozygosity status, either S or Z allele, and MM genotype status.

### 2.5. Statistical Analysis

Qualitative variables were summarized using absolute and relative (percentage) frequencies. Quantitative variables were summarized with means (standard deviations, SD) and medians (interquartile ranges, IQR) depending on a normal and abnormal distribution. Qualitative variables were compared using chi-squared and Fisher exact tests, when appropriate. ANOVAs and Kruskal–Wallis test were used to compare quantitative variables with normal and abnormal distribution. Sidak correlation was adopted for multiple comparisons. A two-tailed *p*-value less than 0.05 was considered statistically significant. We performed statistical computations using IBM SPSS Statistics for Windows version 22.0.

## 3. Results

### 3.1. Study Population and Primary Endpoint

Among 173 CF patients included in the study (57.8% male, median age 27 years (IQR 22.5–34)), prevalence of AAT deficiency was 0. Characteristics of the study population are summarized in Table 1.

### 3.2. Secondary Endpoints

The mean value of AAT was 140 mcg/dL (IQR 126–150.5). Genotype analysis was piMM in 166 (94.9%) patients and piMS in 9 patients (5.1%), respectively. No deficient genotypes were detected in our population. No patients with MS genotype or MM genotype showed low serum levels of AAT. Baseline characteristics of patients with piMM genotype and piMS genotype are summarized in Table 2. No differences were found in terms of demographic characteristics, severity of disease, or concomitant conditions. However, S-allele heterozygosity resulted in significantly lower levels of mean AAT levels than piMM genotype (123 mcg/dL [IQR 108–137.5] versus 140 mcg/dL [IQR 127-153], *p* = 0.017). Analyzing the control population, genotype was piMM in 2667 (93.6%) individuals, piMS in 148 individuals (5.2%), and piMZ in 32 individuals (1.1%). Only one individual in the control population was diagnosed with a deficient genotype (piSS).

### 3.3. Control Population

No differences in terms of genotype characterization were found between the CF population and the control population (Table 3). Recent guidelines and consensus documents suggest that adults with AAT serum less than or equal to 110 mg/dL should undergo SERPIN-1 genetic testing [11,12]. In our cohort population, 12 (6.9%) of CF patients should be genetically tested according to AAT serum level less or equal to 110 mg/dL: 2 (22.2%) in the S-allele heterozygosity group and 10 (6.1%) in the wild-type group (*p* = 0.065).

## 4. Discussion

In consideration of the pathophysiology of CF lung involvement, the screening of CF patients for other diseases enhancing neutrophil inflammation in lungs, as in the case of AAT deficiency, might be considered to break the vicious vortex leading to worse clinical outcomes [3]. The rationale for this approach is to address AAT deficiency not only as a genetic disease but as a mechanism underlying different chronic respiratory conditions [13]. Moving from this consideration, we investigated our cohort with both expanded genotyping and serum AAT levels. However, this report demonstrates that AAT deficiency is not common among patients with CF. To our knowledge, this is the first report of prevalence of AAT deficiency among CF patients coming from a Southern European cohort. Our results confirm a previous experiment from a cornerstone large multicenter study based in Canada, where Frangolias et al. demonstrated the most common AAT-deficient alleles with similar prevalence in both patients with CF and the general population [14].

In our understanding, a partial reduction in AAT levels in the context of a hyper-neutrophilic condition, such as CF lung disease, might behave as a relative deficiency [13]. Thus, we provided a detailed clinical characterization of patients heterozygous for AAT-deficient alleles. Despite CF patients with S allele showing lower levels of AAT serum protein when compared to the piMM genotype, no differences were found between the groups in terms of disease severity and pulmonary function.

Our results question previous controversial associations from pivotal studies, where patients with both CF and piMS and piMZ status showed worse rates of chronic pseudomonal infection in one case or better lung function than the wild-type individuals in the other [15,16]. The results of our study are in line with a previous report from de Faria et al., showing that no significant differences were found in CF clinical severity in patients heterozygous for S or Z alleles compared to the piMM genotype [17].

Moreover, rapidly progressive liver disease has been observed in cases of combination of AAT deficiency and CF [18]. In our study, no significant differences were found in patients with piMS status compared to patients with piMM genotype.

Our study has strengths and limitations. We applied a strong definition of AAT deficiency by the extensive evaluation of both serum and genetic AAT tests. The use of a control group to confirm the results also strengthened our methodology. However, there are some limitations that should be taken into account. First, this was a single-center study conducted in an adult cohort of CF patients. Second, no data concerning the inflammation burden were assessed, as in the case of neutrophil elastase (NE) and other serine proteases [19]. Indeed, AAT augmentation therapy has been found to be effective in reducing the level of serine proteases in both the airway and circulation, reducing elastin degradation, and diminishing airway inflammation [20]. The potential anti-inflammatory and antiapoptotic properties have led to the speculative use of AAT augmentation therapy in a range of both pulmonary and systemic conditions [21]. A recent case report showed a successful administration of intravenous AAT for severe cytokinetic COVID-19 complicated by ARDS in a CF patient [22]. In this case report, rapid decreases in inflammatory parameters were observed following each AAT dose. These were matched by marked clinical and radiographic improvement.

## 5. Conclusions

CF represents one of the most striking examples of a neutrophil-dominated lung inflammation. Screening for other diseases causing lung neutrophil inflammation leading to bronchiectasis, such as AAT deficiency, might be considered to break the vicious vortex leading to worse clinical outcomes. Despite data from our study showing that AAT deficiency is not common among adults with CF, patients with S-allele heterozygosity showed lower levels of AAT serum protein compared to the piMM genotype. However, a relative deficiency in AAT levels in the context of neutrophilic lung inflammation in CF cannot be ruled out [23]. The use of nebulized AAT might be able to overcome some of the shortcomings of intravenous therapy, and it permits delivery directly to the local site of inflammation [24]. To explore this model, further translational research might take into account both AAT and NE concentrations in CF sputum and their balance in the context of a proof-of-concept trial with nebulized AAT treatment in this population.

## Figures and Tables

**Table 1 biomedicines-10-03248-t001:** Demographics, clinical, functional and microbiological characteristics of the study population.

Variables	Study Population (*n* = 173)
*Cystic fibrosis mutation*	
ΔF508S homozygosity (%)	22 (12.7)
At least one residual function mutation (%)	50 (28.9)
*Demographics*	
Female sex, *n* (%)	73 (42.2)
Age, median (IQR)	27 (22.5–34)
BMI, median (IQR)	22.3 (20.4–24)
Underweight, *n* (%)	6 (3.5)
Former or current smoker, *n* (%)	25 (14.5)
*Comorbidities*	
GERD, *n* (%)	22 (12.7)
Nasal polyposis, *n* (%)	51 (29.5)
Chronic sinusitis, *n* (%)	91 (52.6)
Systemic hypertension, *n* (%)	5 (2.9)
Pulmonary hypertension, *n* (%)	7 (4)
Asthma, *n* (%)	11 (6.4)
Osteoporosis, *n* (%)	11 (5.8)
Osteopenia, *n* (%)	26 (13.7)
Depression, *n* (%)	8 (4.6)
Anxiety, *n* (%)	9 (5.2)
History of neoplastic disease, *n* (%)	5 (2.9)
Diabetes, *n* (%)	24 (13.9)
Pancreatic insufficiency, *n* (%)	69 (39.9)
Liver steatosis, *n* (%)	48 (27.7)
Liver cirrhosis, *n* (%)	5 (2.9)
Cholelithiasis, *n* (%)	21 (12.1)
Nephrolithiasis, *n* (%)	13 (7.5)
ABPA, *n* (%)	17 (9.8)
*Functional evaluation*	
FEV1, median (IQR)	82 (63.5–97)
FEV1 < 80%, *n* (%)	77 (44.5)
FEV1 < 50%, *n* (%)	14 (8.1)
FVC, median (IQR)	94 (81–103.3)
*Microbiology*	
Chronic respiratory infection, *n* (%)	132 (76.3)
Chronic P. aeruginosa infection, *n* (%)	83 (48)
MSSA chronic infection, *n* (%)	65 (37.6)
*Clinical status*	
Exacerbations, median (IQR)	1 (0–3)
Exacerbations ≥2 previous year, *n* (%)	79 (45.7)
Exacerbations ≥3 previous year, *n* (%)	53 (30.6)
Hospital admission 1+, *n* (%)	52 (30.1)
Total antibiotic courses per year, median (IQR)	2 (1–3)
LTOT, *n* (%)	3 (1.7)
Daily sputum, *n* (%)	136 (78.6)
Sputum volume, median mL (IQR)	15 (5–35)
*Chronic treatment*	
Chronic macrolide therapy, *n* (%)	32 (18.5)
Chronic antibiotic inhaled therapy, *n* (%)	63 (36.4)
Antifungal, *n* (%)	22 (12.7)
Respiratory physiotherapy, *n* (%)	132 (76.3)

BMI: body mass index; IQR: interquartile range; GERD: gastroesophageal reflux disease; ABPA: allergic bronchopulmonary aspergillosis; FEV1: forced expiratory volume in the first second; FVC: forced vital capacity; LTOT: long-term oxygen therapy; MSSA: Methicillin-susceptible *Staphylococcus aureus*.

**Table 2 biomedicines-10-03248-t002:** Comparison of demographics and clinical, functional and microbiological characteristics of the two study groups.

Variables	MM (*n* = 164)	MS (*n* = 9)	*p* Value
*Cystic fibrosis mutation*			
ΔF508S homozygosity (%)	21 (12.8)	1 (11.1)	0.882
At least one residual function mutation (%)	46 (28)	4 (44.4)	0.291
*Demographics*			
Female sex, *n* (%)	70 (42.7)	3 (33.3)	0.580
Age, median (IQR)	27 (22.3–34)	30 (23–33)	0.907
BMI, median (IQR)	22.2 (20.4–23.9)	23.7 (20.1–26.7)	0.380
Underweight, *n* (%)	5 (3.1)	1 (11.1)	0.203
GERD, *n* (%)	19 (11.6)	3 (33.3)	0.057
Former or current smoker, *n* (%)	23 (14)	2 (22.2)	0.496
*Comorbidities*			
Nasal polyposis, *n* (%)	50 (30.5)	1 (11.1)	0.214
Chronic sinusitis, *n* (%)	88 (53.7)	3 (33.3)	0.234
Systemic hypertension, *n* (%)	5 (3)	0	0.595
Pulmonary hypertension, *n* (%)	7 (4.3)	0	0.527
Asthma, *n* (%)	11 (6.7)	0	0.422
Depression, *n* (%)	8 (4.9)	0	0.497
Anxiety, *n* (%)	9 (5.5)	0	0.470
History of neoplastic disease, *n* (%)	5 (3)	0	0.595
Diabetes, *n* (%)	22 (13.4)	2 (22.2)	0.677
Pancreatic insufficiency, *n* (%)	66 (40.2)	3 (33.3)	0.680
Liver steatosis, *n* (%)	44 (26.8)	4 (44.4)	0.250
Liver cirrhosis, *n* (%)	5 (3)	0	0.868
Cholelithiasis, *n* (%)	18 (11)	0	0.519
Nephrolithiasis, *n* (%)	12 (7.3)	1 (11.1)	0.674
ABPA, *n* (%)	16 (9.8)	1 (11.1)	0.894
*Functional evaluation*			
FEV1, median (IQR)	82 (65–97)	73 (49.5–101.5)	0.659
FEV1 < 80%, *n* (%)	72 (46.2)	5 (55.6)	0.583
FEV1 < 50%, *n* (%)	12 (7.7)	2 (22.2)	0.128
FVC, median (IQR)	94 (81.5–103.5)	94 (67.5–94)	0.643
*Microbiology*			
Chronic respiratory infection, *n* (%)	125 (76.2)	7 (77.8)	0.915
Chronic *P. aeruginosa* infection, *n* (%)	77 (47)	6 (66.7)	0.249
Chronic MSSA infection, *n* (%)	54 (32.9)	3 (33.3)	0.629
*Clinical status*			
Exacerbations, median (IQR)	1 (0–3)	1 (0.5–2)	0.629
Exacerbations ≥2 previous year, *n* (%)	76 (46.3)	3 (33.3)	0.446
Exacerbations ≥3 previous year, *n* (%)	52 (31.7)	1 (11.1)	0.192
Hospital admission 1+, *n* (%)	51 (31.1)	1 (11.1)	0.203
Total antibiotic courses per year, median (IQR)	2 (1–3)	2.5 (1.5–3.5)	0.572
LTOT, *n* (%)	3 (1.8)	0	0.682
Daily sputum, *n* (%)	130 (79.3)	6 (66.7)	0.369
Sputum volume, median mL (IQR)	15 (6.3–30)	22.5 (5–50)	0.896
**Chronic treatment**			
Chronic macrolide therapy, *n* (%)	30 (18.3)	2 (22.2)	0.768
Chronic antibiotic inhaled therapy, *n* (%)	60 (36.6)	3 (33.3)	0.844
Antifungal, *n* (%)	21 (12.8)	1 (11.1)	0.920
Respiratory physiotherapy, *n* (%)	127 (77.4)	6 (66.7)	0.455

BMI: body mass index; IQR: interquartile range; GERD: gastroesophageal reflux disease; ABPA: allergic bronchopulmonary aspergillosis; FEV1: forced expiratory volume in the first second; FVC: forced vital capacity; LTOT: long-term oxygen therapy; MSSA: methicillin-susceptible *Staphylococcus aureus*.

**Table 3 biomedicines-10-03248-t003:** Study cohort and prevalence of SERPINA1 variants in cases.

Variable	CF Patients	Healthy Controls	*p*-Value
*n*	173	2848	
Age, years	29.1 (8.7)	42.8 (13.2)	<0.001
Male	100 (57.8)	1914 (67.2)	<0.001
Active/former smokers	25 (14.5)	409 (16.7)	0.973
*Genotype* (%)			0.554
PiMM	164 (94.9)	2667 (93.6)	N.A.
PiMS	9 (5.1)	148 (5.2)	N.A.
PiMZ	0 (0)	32 (1.1)	N.A.
PiSS	0 (0)	1 (0.0)	N.A.
PiZZ	0 (0)	0 (0.0)	N.A.

Data are presented as mean (SD) and *n* (%). N.A.: not applicable.

## Data Availability

The data presented in this study are available on request from the corresponding author. The data are not publicly available due to privacy and ethical restrictions.

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
