# Peer review of "Genetic and Serum Screening for Alpha-1-Antitrypsin Deficiency in Adult Patients with Cystic Fibrosis: A Single-Center Experience"

_biomedicines, 2022, doi:10.3390/biomedicines10123248_

Round 1

Reviewer 1 Report

The authors should further improve their manuscript, particularly language and style. However, this study does not hold much of the novelty- there are many previous studies in this subject, demonstrating AAT deficiency does not correlate with severity of cystic fibrosis.

Author Response

Comment 1:The authors should further improve their manuscript, particularly language and style.

Response to comment 1: We thank the reviewer for her/his comment and we agree with her/him.We improve language and style of our manuscript (see track changes).

Comment 2: However, this study does not hold much of the novelty- there are many previous studies in this subject, demonstrating AAT deficiency does not correlate with severity of cystic fibrosis.

Response to comment 2: We thank the reviewer for her/his comment and we recognize that previous studies on this subject have been published. However, this is the first study in a Southern European cohort of patients. To clarify the importance of our results and the great topical interest we decided to improve the discussion section.

Reviewer 2 Report

biomedicines-2074885

Genetic and serum screening for alpha-1-antitrypsin deficiency 2 in adult patients with cystic fibrosis: a single center experience

The article is distinguished by the originality of the selected topic, structured correctly and written in Standard English. The manuscript presents the prospective study aimed to investigate the prevalence of alpha-1 antitrypsin deficiency (AATD) in adults with Cystic Fibrosis (CF), including adults with cystic fibrosis, for the period January 2018 to March 2019, conducted in Milan.

 The importance of the study: 1)AATD is not common among adults with CF; 2)patients with S-allele heterozygosity showed lower levels of AAT serum protein compared to the piMM genotype; 3)the hypothesis of a relative deficiency of AAT levels in the context of neutrophilic lung inflammation in CF; 4) further translational research might take into account both AAT and NE concentrations in CF sputum and their balance in the context of a proof-of-concept trial with nebulized AAT treatment in this population.

The introduction, methods, and results are presented correctly and a logical relationship between them is clearly observed. In an initial review of the article there were stylistic and grammatical mistakes that I hope the authors will avoid after revision.

 Suggest:

1.      Most of the literature used is before 2020! I suggest it be updated! I suggest adding at least 10 more literature sources to validate the results achieved.

2.      The discussion part is small. I suggest that the discussion be expanded further and supported with materials from the last two 21-22 years.

3.      I have no objections to the table's design! 

Author Response

General comment: The article is distinguished by the originality of the selected topic, structured correctly and written in Standard English. The manuscript presents the prospective study aimed to investigate the prevalence of alpha-1 antitrypsin deficiency (AATD) in adults with Cystic Fibrosis (CF), including adults with cystic fibrosis, for the period January 2018 to March 2019, conducted in Milan. The importance of the study: 1)AATD is not common among adults with CF; 2)patients with S-allele heterozygosity showed lower levels of AAT serum protein compared to the piMM genotype; 3)the hypothesis of a relative deficiency of AAT levels in the context of neutrophilic lung inflammation in CF; 4) further translational research might take into account both AAT and NE concentrations in CF sputum and their balance in the context of a proof-of-concept trial with nebulized AAT treatment in this population.The introduction, methods, and results are presented correctly and a logical relationship between them is clearly observed.In an initial review of the article there were stylistic and grammatical mistakes that I hope the authors will avoid after revision.

Response to general comment: We would like to thank the reviewer for her/his nice words. We review the text fixing grammatical mistakes.

Comment 1: Most of the literature used is before 2020! I suggest it be updated! I suggest adding at least 10 more literature sources to validate the results achieved. 

Response to comment 1: We thank the reviewer for her/his comment and we agree with her/him. However few studies have been published concerning this topic in the last years. We update the text and references with recent data.

Comment 2: The discussion part is small. I suggest that the discussion be expanded further and supported with materials from the last two 21-22 years.

Response to comment 2: We thank the reviewer for her/his comment and we agree with her/him. We expand the discussion with recent data as follows "

Our results questioned previous controversial associations from pivotal studies, where patients with both CF and piMS and piMZ status showed worse rates of chronic pseudomonal infection in one case or better lung function than the wild-type individuals in the other [15,16]. The results of our study are in line with previous reports from de Faria et al, showing that no significant differences were found in CF clinical severity in patients heterozygous for S or Z alleles compared to the piMM genotype [17].

Moreover, rapidly progressive liver disease has been observed in case of combination of AAT deficiency and CF [18]. In our study, no significant differences were found in patients with piMS status compared to patients with piMM genotype.

Our study has strengths and limitations. We applied a strong definition of AAT deficiency by the extensive evaluation of both serum and genetic AAT tests. The use of a control group to confirm the results also strengthened our methodology. However, there are some limitations that should be taken into account. First, this is a single center study conducted in an adult cohort of CF patients. Second, no data concerning the inflammation burden was assessed, as in the case of neutrophil elastase (NE) and other serine proteases [19]. Indeed, AAT augmentation therapy has been found to be effective at reducing the level of serine proteases in both the airway and circulation, reducing elastin degradation, and diminishing airway inflammation [20]. The potential anti-inflammatory and antiapoptotic properties, have led to the speculative use of AAT augmentation therapy in a range of both pulmonary and systemic conditions [21]. A recent case report has showed a successful administration of intravenous AAT for severe cytokinemic COVID-19 complicated by ARDS in a CF patient [22]. In this case report, rapid decreases in inflammatory parameters were observed following each AAT dose. These were matched by marked clinical and radiographic improvement."

Round 2

Reviewer 1 Report

The manuscript needs further language corrections. Sentences are too long and do not always have a proper logical transition. I leave a final decision to the Editor.